# Nano Silica and Metakaolin Effects on the Behavior of Concrete Containing Rubber Crumbs

**Navid Chalangaran** [1] , **Alireza Farzampour** [2,]* **and Nima Paslar** [1]

1   Department of Civil Engineering, Qeshm Branch, Islamic Azad University, Qeshm, P.O. Box 7915893144,
    Iran; Chalangaran@Gmail.com (N.C.); n.paslar@iauqeshm.ac.ir (N.P.)
2   Department of Civil and Environmental Engineering, Virginia Tech, Blacksburg, VA 24060, USA
*   Correspondence: afarzam@vt.edu

**Abstract:** The excessive production of worn tires remaining from the transportation system and the lack of proper procedures to recycle or reuse these materials have caused critical environmental issues. Due to the rubber's toughness, this material could be implemented to increase concrete toughness, and by crushing the tires concrete aggregates can be replaced proportionally with rubber crumbs and large quantities of scrapped rubber. However, this substitution decreases the concrete strength. In this study, crushed rubber with sizes from 1 to 3 mm and 3 to 6 mm were replaced by 5%, 10%, and 15% sand; the combination of two additives of nano silica and metakaolin additives with optimum values was used to compensate the degradation of the strength and improve the workability of the concrete. Moreover, the compressive strength, tensile behavior, and modulus of elasticity were measured and compared. The results indicate that the optimum use of nano silica and metakaolin additives could compensate the negative effects of the rubber material implementation in the concrete mixture while improving the overall workability and flowability of the concrete mixture.

**Keywords:** nano silica additive; metakaolin additive; rubber crumbs; recyclable rubber; concrete strength

## 1. Introduction

Concrete is typically implemented to have a high compressive and tensile strength for use in various structural applications. However, this material loses its resistance under harsh environmental conditions, causing crack propagation and degradation. There are several producers to improve concrete behavior against crack initiation with the use of additive material, fibers, and rubber crumbs with different shapes and mechanical properties, leading to a desirable resistance and durability [1]. It was previously shown that adding blast furnace slag and fly ash could be useful in developing concrete resistance. However, the use of mineral admixtures is limited, and it is necessary to study improvement methods for general applications [2,3].

Numerous studies have been conducted on the use of recycled materials in concrete. [4–7]. Each year, more than 242 million worn tires are discarded in the United States, which account for more than 1.2% of the total weight of national waste [8]. In addition to urban rubber waste, many billions of tires are buried or burned in valleys, which has several disadvantageous effects on human health and causes significant environmental pollution [9]. The use of rubber crumbs in the concrete asphalt mixture increases the plasticity of the whole mixture and protects the cured concrete from cracking and shrinkage under thermal expansion and contraction. In recent years, extensive researchers have performed studies on the use and application of rubber crumbs in concrete [3]. Studies on the rubber size effect on concrete strength properties have shown that, although the strength of the concrete containing rubber crumbs decreases compared to the mixtures made of aggregates, the plastic capacity of the concrete increases considerably. Replacing a part of the sand with rubber ash decreases

the air in the concrete mixture, increases the setting time, decreases the compressive and flexural strength, and leads to desirable resistance against freeze-thaw cycles [10]. It is shown that using low volumes of rubber in a concrete mixture reduces the risk of the sudden crusting of the rubber at high temperatures [11].

The replacement of rubber instead of sand causes possible initial crack propagation in the concrete, reduces the percentage of the total crack area, and increases the flexibility of the concrete [12]. Rubber granules with rough surfaces or treated with water or acid show appropriate adhesion to concrete, leading to a high compressive strength [13]. Along the same lines, replacing rubber crumbs with fine aggregate decreases the compressive strength by up to 65% of the original strength, and by using rubber crumbs instead of the coarse aggregate the compressive strength could be reduced by more than 85% [14], which necessitates the need for further investigations to improve the concrete mixture. Along the same lines, previous studies have shown that no considerable changes have been reported in compressive strength development for concrete mixtures with rubber crumbs with a size of 4.75 mm or less and with 1% to 10% weight of cement to concrete. One of the factors that reduces the strength of concrete containing rubber crumbs is having smaller holes filled with water causing cavities [15]. Concrete containing coarse rubber crumbs could withstand high pressures after failure, undergoing large shifts without being disjointed from the rest of the structure; hence, the occurring displacements and deformations are reversible after unloading [16,17]. Researchers have previously conducted several detailed studies on the crumb rubber effects in concrete. Research has shown that concrete containing rubber crumbs performs better in chloride permeability and electrical insulator increasing, but the concrete develops lower strength properties. For this purpose, several studies are conducted to solve the problem of decreasing the strength of concrete by using silica foam, reinforcement and other concrete additive materials. The high price of some of the consumables used or the lack of change in lubricants and the flow ability of the concrete could develop issues in various cases [18–21].

In addition, the use of a metakaolin additive in concrete is associated with noticeable improvements in the concrete ultimate compressive strength. The replacement of cement with 5% to 15% of metakaolin additive could improve the concrete compressive strength up to 45% on average [22–27]. Recent studies have investigated the effect of using a combination of nano silica and metakaolin in concrete simultaneously. In this study, the combination of 1% nano-silica and 10% metakaolin, which replaced part of the cement, had the best results in the mechanical properties of concrete, with a 15% increase in compressive strength and 40% increase in the flexural and tensile strength of concrete [28]. Along the same lines, the implementation of the nano silica additive material to improve the concrete strength containing rubber was associated with negative effects such as the dehydration and crushing of the fresh concrete. The nano silica concrete shrinkage rate is large, causing early dry shrinkage and affecting the overall strength. For this purpose, the effects of nano silica and metakaolin additives with optimum values of the crushed rubber crumbs with the size of 1 to 3 mm and 3 to 6 mm are investigated in this study. The compressive strength, tensile behavior, and modulus of the elasticity were measured and procedures to improve the strength development of the concrete containing recyclable rubber crumbs are investigated and elaborated in detail.

## 2. Materials and Methods

The mix design is designed according to ACI 211-89 [29], for which the slump is considered tp be between 50 and 70 mm. The compressive strength is measured based on samples with dimensions of $15 \times 15$ cm for 7, 14, and 28 curing days. The splitting tensile strength and Modulus of elasticity are evaluated with $15 \times 30$ cm cylinder samples. Each additive is examined separately for three different alternatives for strength development. Ultimately, the optimal values for Nano-silica with Metakaolin additives are obtained and combined with the maximum weight amount of rubber crumbs.

The cement used in this study is chloride resistant TYPE 2 Fars No for having a lower hydration heat. In the binary phase diagrams, the combination of two phases is located on the x-axis and the temperature on the y-axis. When the experimental work is completed, the various mineral phases

obtained by mixing the two oxides are precisely determined, and the melting and freezing temperatures of these compounds are developed. Between the melting and freezing temperatures of these materials, some crystals are clearly formed in the liquid phase. It is observed that, at temperatures close to freezing temperature, more crystals form inside of the material. The $CaO-SiO_2$ fuzzy diagram shows that the minimum temperature at which CaO and $SiO_2$ can melt is 2050 °C. The minimum temperature at which C3S and C2S can coexist is 1450 °C. At a temperature of 500 to 600 °C, water comes out of the clay material. At a temperature of 900 °C, limestone becomes carbonated and at a temperature of 1100 °C, cristobalite melts, and its combination with lime intensifies. At 1200 °C, there was no form of free silica.

It is noted that the cement used in this study is chloride-resistant TYPE 2 Fars No, for having lower hydration heat. A chemical analysis of this cement is provided in Table 1. The rubber crumbs used in this research were provided from the tire wastes of cars with two different sizes. Crushed rubbers with dimensions of 1 to 3 mm are named "PR" and the ones with dimensions of 3 to 6 mm are named "CR" accordingly (Table 2). In addition, the "om.on" samples show the optimal amount of nano silica and metakaolin additives combined with different percentages of the rubber material. In addition, the nano-silica used in this study loses 9.5% of its weight after one hour at 900 °C and melts at 1700 °C.

**Table 1.** Common properties of the concrete chemical content.

| | Chemical Component (%) | | | | | | | | |
|---|---|---|---|---|---|---|---|---|---|
| | **$SiO_2$** | **$Al_2O_3$** | **$Fe_2O_3$** | **CaO** | **MgO** | **$SO_3$** | **$K_2O$** | **$Na_2O$** | **L.O.I** |
| Cement | 21 | 5.4 | 4.21 | 63.59 | 1.7 | 1.8 | 0.8 | 0.12 | 1.38 |
| Metakaolin | 51.85 | 43.87 | 0.99 | 0.2 | 0.18 | - | 0.12 | 0.01 | 0.57 |
| Nano-Silica | 98 | - | 0.294 | 0.393 | - | 0.185 | - | 0.328 | - |

**Table 2.** Mixture proportions of the concrete (units are based on $kg/m^3$).

| Sample Name | Cement ($kg/m^3$) | Sand ($kg/m^3$) | Coarse Aggregate ($kg/m^3$) | Water ($kg/m^3$) | Nano Silica ($kg/m^3$) | Metakaolin ($kg/m^3$) | Rubber ($kg/m^3$) | Super Plasticizer ($kg/m^3$) | Slump (mm) |
|---|---|---|---|---|---|---|---|---|---|
| Control | 462.2 | 732.39 | 992 | 133.52 | - | - | - | 1.39 | 60 |
| N 1% | 457.5 | 732.39 | 992 | 133.52 | 4.62 | - | - | 1.85 | 45 |
| N 3% | 448.3 | 732.39 | 992 | 133.52 | 13.86 | - | - | 2.31 | 30 |
| N 5% | 439.09 | 732.39 | 992 | 133.52 | 23.11 | - | - | 2.8 | 39 |
| M 5% | 439.09 | 732.39 | 992 | 133.52 | - | 23.11 | - | 1.59 | 62 |
| M 10% | 415.98 | 732.39 | 992 | 133.52 | - | 46.22 | - | 1.85 | 60 |
| M 15% | 392.8 | 732.39 | 992 | 133.52 | - | 69.3 | - | 2.1 | 45 |
| Pr 5% | 462.2 | 695.79 | 992 | 133.52 | - | - | 36.6 | 1.39 | 65 |
| Pr 10% | 462.2 | 659.16 | 992 | 133.52 | - | - | 73.23 | 1.59 | 55 |
| Pr 15% | 462.2 | 622.54 | 992 | 133.52 | - | - | 109.85 | 1.65 | 50 |
| Cr 5% | 462.2 | 695.79 | 992 | 133.52 | - | - | 36.6 | 1.39 | 55 |
| Cr 10% | 462.2 | 659.16 | 992 | 133.52 | - | - | 73.23 | 1.39 | 50 |
| Cr 15% | 462.2 | 622.54 | 992 | 133.52 | - | - | 109.85 | 1.39 | 42 |
| om.on. Pr 5% | 346.68 | 695.79 | 992 | 133.52 | 13.86 | 69.3 | 36.6 | 3.2 | 55 |
| om.on. Pr 10% | 346.68 | 659.16 | 992 | 133.52 | 13.86 | 69.3 | 73.23 | 3.2 | 58 |
| om.on. Pr 15% | 346.68 | 622.54 | 992 | 133.52 | 13.86 | 69.3 | 109.85 | 3.2 | 60 |
| om.on. Cr 5% | 346.68 | 695.79 | 992 | 133.52 | 13.86 | 69.3 | 36.6 | 3.2 | 50 |
| om.on. Cr 10% | 346.68 | 659.16 | 992 | 133.52 | 13.86 | 69.3 | 73.23 | 3.2 | 55 |
| om.on. Cr 15% | 346.68 | 622.54 | 992 | 133.52 | 13.86 | 69.3 | 109.85 | 3.2 | 60 |

Waste rubber crumbs generally contain wire and fabric, and for the best results these added materials were initially washed and separated from the rubber. The sand used in this experiment was mineral with a dry weight of 1590 $kg/m^3$ and maximum dimensions of 4.75 mm, for which the aggregation graph is shown in Figure 1. The gravel aggregates are used with dry weight of 1620 $kg/m^3$ and the maximum dimension of 12.5 mm. The almond coarse gravel are considered to have a dry weight of 1600 $kg/m^3$ and maximum dimensions of 19 mm. Super plasticizer with commercial brand of Super Plast P.C 5000N is used to uniformly spread the concrete and prevent the particles from sticking together again by obstructing the space between particles. This also decreases the mix water up to 30% to improve the ultimate concrete strength.

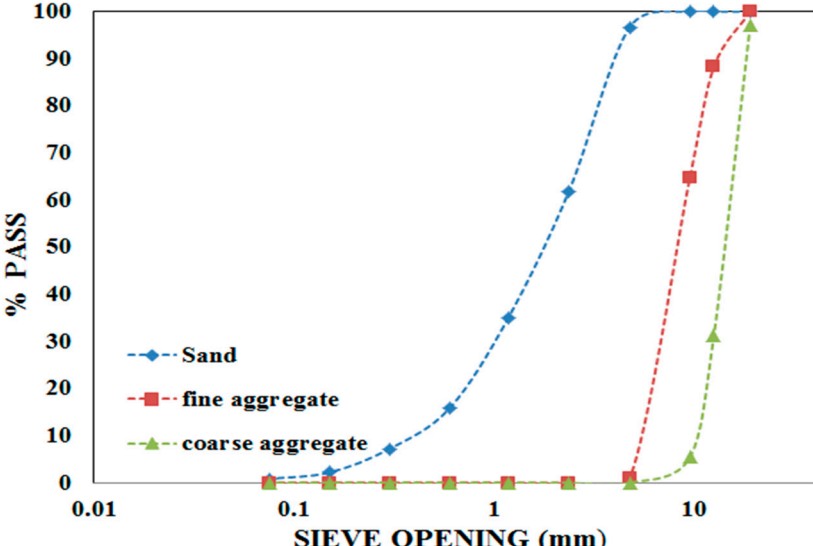

**Figure 1.** Grading curves for the used aggregates in concrete mixtures.

The mix design was developed based on the standard procedure of ACI 211-89 [24]. For this purpose, four different mix designs with the ultimate strength of 400 kg/cm$^2$ were considered, and each mix design included three different alternatives with a specific additive. The water–cement ratio was considered to be 0.29, and the slump was kept in the range of 50 ± 20 mm. Super plasticizer was initially mixed with water and poured into the mixer, then additives (nano silica or metakaolin) were added to the mix. Subsequently, the aggregates and cement were poured into the mixer. For the sample with rubber crumbs, first the rubber crumbs and sand were mixed and then poured into the mixture following the producers for the rest of the mix designs.

After thorough mixing, the concrete slump was specified and then sampling was performed. It is noted that in all the sampling processes the casts were filled in three layers and each layer was tapped 25 times and finally smoothed and kept in a maintenance pool following the ASTM [25] procedures. Due to the high-water absorption of nano silica and metakaolin additives, the super plasticizer was added to the mixture for having a desirable uniformly. Table 2 summarizes the samples considered for further investigating the nano silica and metakaolin effects on the strength development of concrete with recyclable rubber crumbs.

The compressive strength, splitting tensile strength, and modulus of elasticity were assessed according to the ASTM C469 and C39 [25] standards, which are shown in Figure 2. To conduct the compressive strength test, nine cubes with dimensions of 15 × 15 cm following the standard procedures with ages of 28, 14, and 7 days were selected and evaluated. In addition, for the tensile test, three samples with 15 × 30 cm dimensions and 28 cured days were considered for each mix design. Along the same lines, to test static modulus of elasticity using module enclosure, three samples with 15 × 30 cm dimensions and a curing age of 28 days were assessed.

After drying the samples, to measure the compressive strength of the concrete samples were placed on two flat surfaces for further investigation. The constant loading speed with an increasing rate of 0.4 MPa/s was applied to the samples, and by dividing the ultimate force applied to the cross-sectional area the concrete compressive strength was accordingly calculated. In Figure 2, to show the device and used equipment, samples containing crumb rubber to measure compressive strength (a), tensile strength (b) and modulus of elasticity (c) are considered.

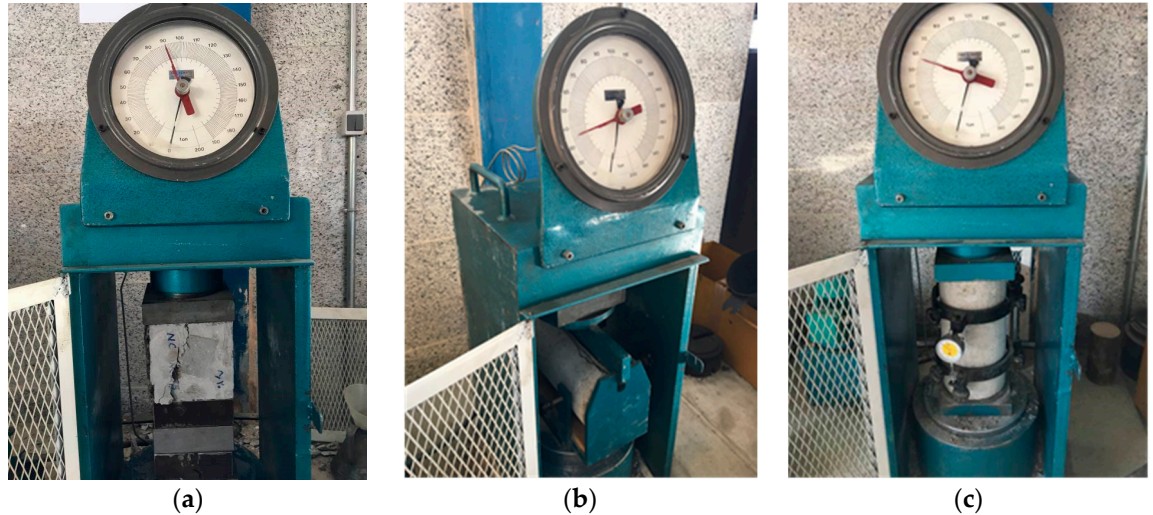

(a)         (b)         (c)

**Figure 2.** Conducted tests on the behavior of the concrete with rubber crumbs. (**a**) Compressive strength test for cubic specimens; (**b**) Splitting tensile strength test for cylinder specimens; (**c**) Modulus of elasticity test for cylinder specimens.

## 3. Results and Discussion

The compressive strengths of various mix designs are assessed and compared according to the ASTM [30] provisions, as is shown in Figures 3–6. Compared to the control sample, the samples containing 3% nano silica have the highest compressive strength development, with a 24% increase in ultimate strength for the 28-day curing condition (Figure 3). Along the same lines, the samples containing 15% metakaolin have the highest compressive strength development and achieved a 34% higher ultimate strength compared to the control sample, which is shown in Figure 4. It is noted that, on average, adding nano silica additive could lead to a 3% higher compressive strength, which is determined from Figures 3 and 4.

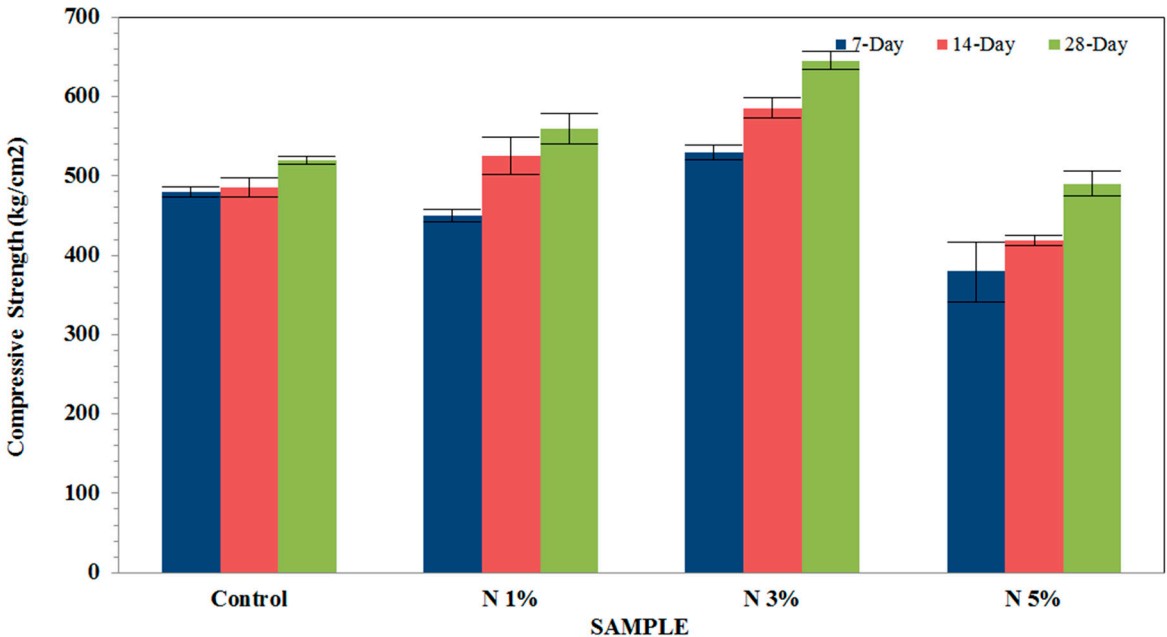

**Figure 3.** Ultimate compressive strength for the mixture with the nano silica additive.

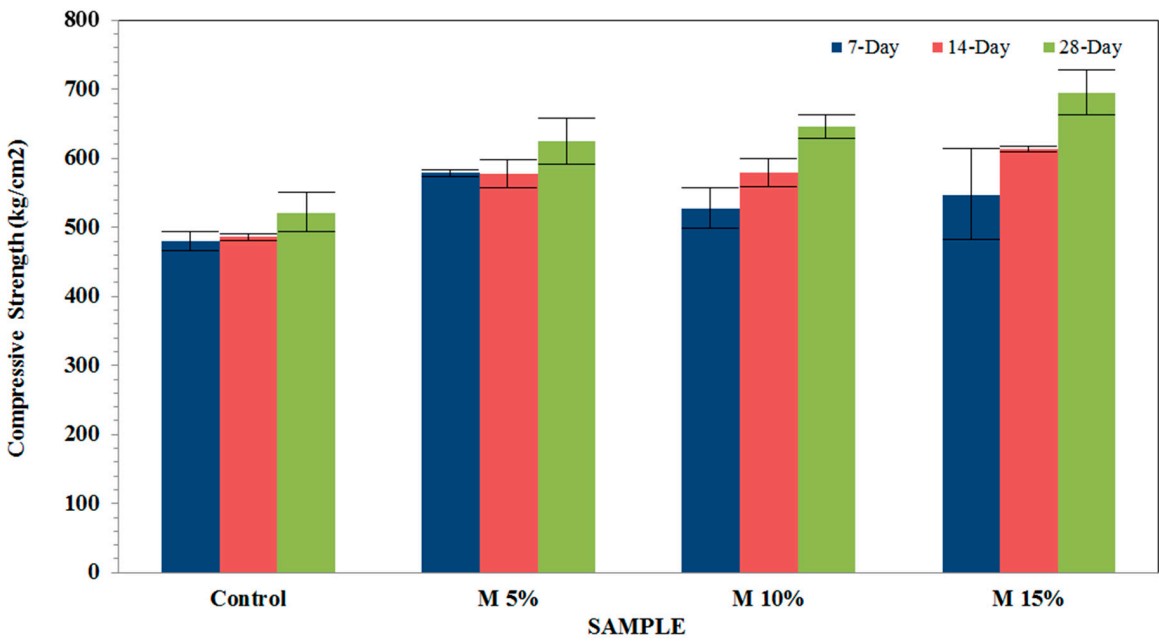

**Figure 4.** Ultimate compressive strength for the mixture with the metakaolin additive.

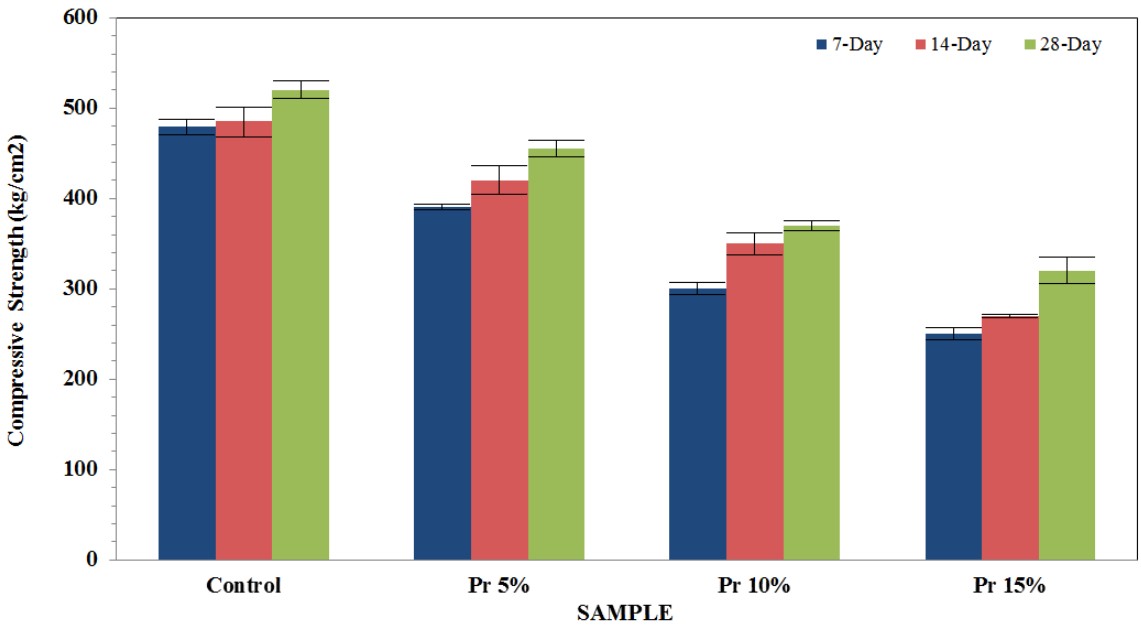

**Figure 5.** Ultimate compressive strength for the mixture with the fine rubber crumb (Pr).

From the samples containing rubber crumbs, samples containing 5% rubber crumbs had the highest strength, indicating that the rubber crumbs could negatively affect the ultimate compressive strength of the concrete. It is noted that by increasing the amount of rubber up to 15%, the compressive strength of the samples decreases up to 38% and 47% of the original sample on average for fine and coarse rubber crumbs, respectively (Figures 5 and 6). It is observed that samples with rubber crumbs were not completely broken under high deformations, resulting in lower crack propagations.

Due to the softer texture of the rubber material compared to aggregates, the cracks were developed around the rubber particles during the loading condition and could propagate within the concrete mixture subsequently. It is shown that the negative effects of crumb rubber implementation on the concrete ultimate strength could be compensated for by the desirable improvements in ductility, resistance to secondary cracking, and the flowability of the concrete mixture. Figure 7 shows the

compressive strength values of the specimens. It is observed that samples with 15% fine-aggregated rubber crumbs, 3% nano silica, and 15% metakaolin additives could achieve the highest ultimate compressive strength difference compared to the rest of the specimens, while having a desirable workability and flowability. In addition, the results show that the rubber particle has a significant reduction in strength due to the lack of adhesion with the concrete. Despite the desirable addition with nano silica and metakaolin, a significant drop in resistance in samples containing 15% rubber was determined. As shown in Figure 7, in case of using fine-grained tire rubber crumbs, the resistance drop can be partially compensated for by the optimal percentage of additives.

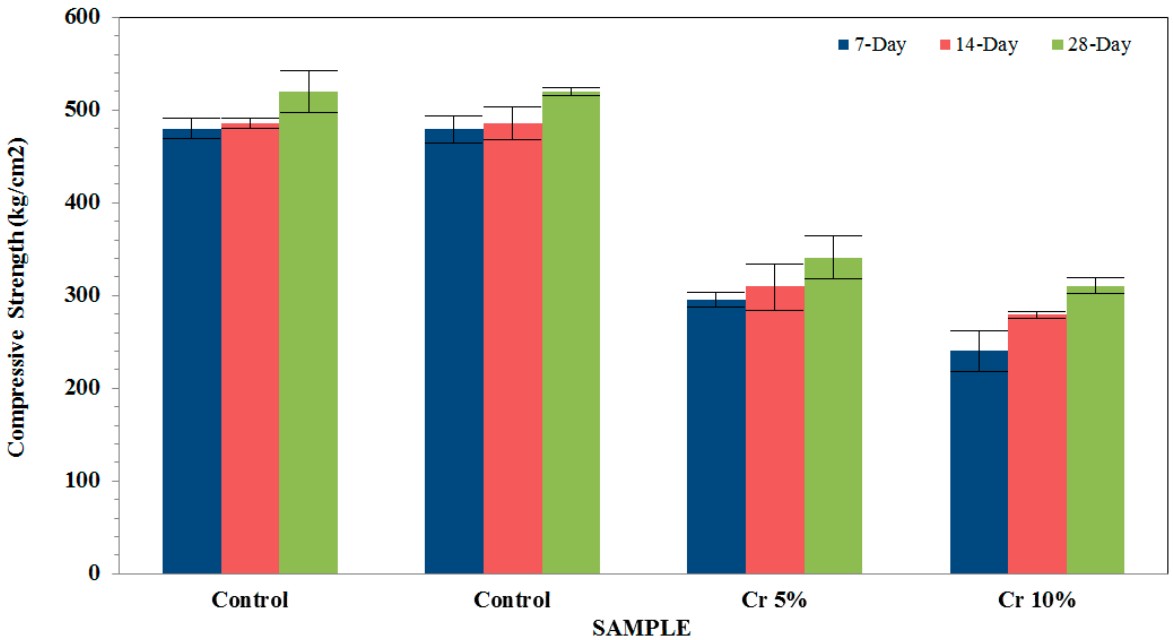

**Figure 6.** Ultimate compressive strength for the mixture with the coarse rubber crumb (Cr).

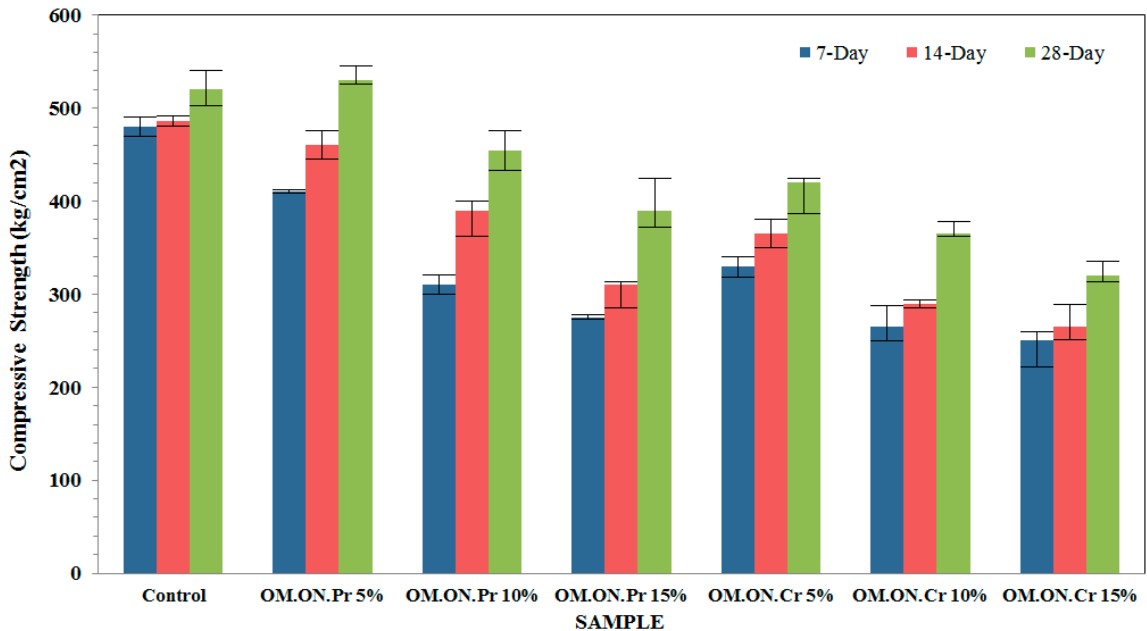

**Figure 7.** Ultimate compressive strength of the samples with rubber crumbs and additives.

## 4. Discussion

As is shown in the Figure 8, the samples containing 15% metakaolin and 3% nano silica could effectively improve the behavior of the concrete for the tensile strength. Although the durability of the concrete has been increased by adding the rubber crumbs, the general tensile strength is reduced by 25% on average. It is concluded that if the samples are designed with the optimum percentages for metakaolin and nano silica additives, the ultimate strength would be reduced by 16% regardless of the rubber crumb size, which eliminates the negative effects of rubber crumbs' implementation in the concrete mixture. It is shown that the use of 10% more rubber crumbs would lead to an 18% reduction in tensile strength, as is determined in Figure 8.

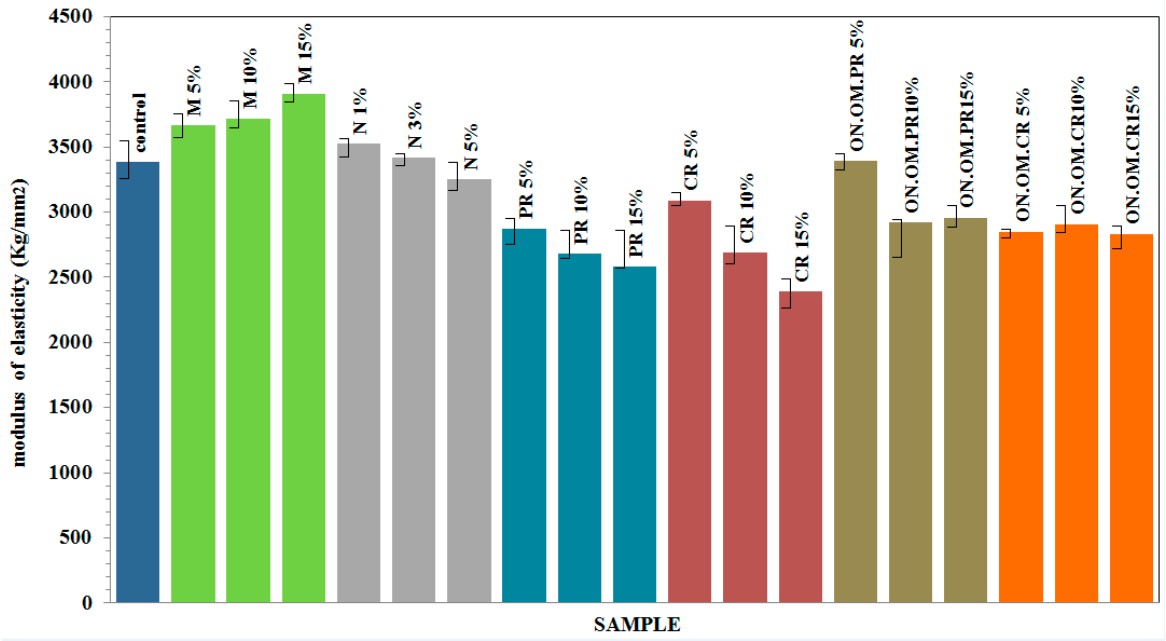

**Figure 8.** Splitting tensile strength evaluation for the samples with rubber crumbs and additives.

The modulus of elasticity is evaluated based on the two provisions of the ASTM C469 [25], which is directly correlated to the ultimate compressive strength, as is shown in Figure 9. It is concluded that samples with a higher ultimate compressive strength have a higher modulus of elasticity. The effect of finer rubber crumbs on the modulus of elasticity is less than the ultimate compressive strength and tensile behavior. In addition, adding five percent metakaolin additive could improve the modulus of elasticity up to 8%, while adding five percent nano silica could lead to a lower modulus of elasticity of up to 4%. By increasing the coarse rubber crumbs, the modulus of elasticity has remained unchanged for the samples with optimum additives; however, adding five more percent fine rubber crumbs leads to a 15% reduction in the modulus of elasticity. It is shown that, within small loads, there is a permanent deformation which makes the elastic modulus not be constant. It is concluded that this test depends on factors such as the material, the type of cement, the number of additives, the loading speed, and the dimensions of the samples. These factors justify the non-uniformity of the results, as shown in Figure 9.

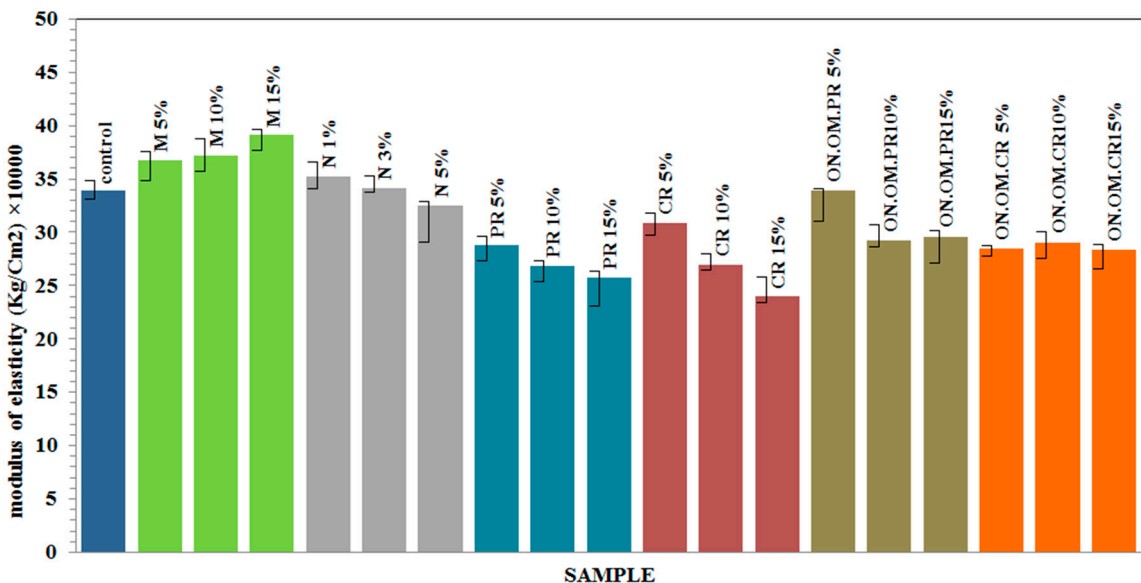

**Figure 9.** Modulus of elasticity for the samples with rubber crumbs and additives.

## 5. Conclusions

Rubber could be implemented for increasing concrete toughness, and by crushing tires concrete aggregates can be replaced proportionally with rubber crumbs and large quantities of scrapped rubber. Concrete-containing rubber crumbs have better integration compared to conventional concrete, leading to proper flowability and durability. Using rubber crumbs in concrete reduces the ultimate compressive strength, splitting the tensile strength and modulus of elasticity of the concrete. Despite the reduction in the ultimate compressive strength of the concrete with rubber crumbs, it could have a softer failure and less deformation compared to conventional concrete, which could negatively affect the coarser rubber crumb. It is shown that, by using the optimum percentage of nano silica and metakaolin additives, the negative effects of rubber crumbs in concrete mix are reduced. It is also determined that by using nano silica and metakaolin additives in the concrete mix design, the water ratio should be set 0.4 or more to compensate for the high water absorption of the additives.

Using rubber in concrete could generally change the concrete failure. By increasing the rubber content used in concrete, a softer failure would be observed and the concrete would not yield to failure after reaching the ultimate load with less cracking. This behavior represents the capability of the high-energy absorption and endurance of the concrete containing compressed rubber crumbs. However, due to the mixing issues of rubber crumbs in the concrete mixture, it is recommended to use this material for replacement purposes with less than 5% of the concrete total weight, causing a desirable distribution throughout the concrete matrix. In addition, it is recommended to use washed finer rubber granules without sharp corners to reduce the negative effects on the workability of the concrete.

Concrete containing crumbs of rubber has more ductile breaking patterns under compressive loads and could withstand loads of at least half its original capacity. This performance of concrete containing crumbs seems to be desirable for several operations on road obstacles.

In recent decades, there has been a significant increase in recycling industrial tube materials and car tires. The tires are either buried in pits or set on fire, producing significant environmental pollution. In the present study, due to the acceptable strength of the designed concrete, it is possible to use this type of concrete for production in highway New Jersey blocks, heavy wall blocks, partition walls, and other general concreting. Thus, crumb rubber could be used as an aggregate without polluting the environment and with a comparable high mechanical strength.

**Author Contributions:** Conceptualization, N.C. and A.F.; methodology, N.C.; software, N.C. and N.P.; validation, N.C., A.F., and N.P.; formal analysis. N.C.; investigation, N.C.; data curation, N.C.; writing—original draft preparation, N.C., A.F., and N.P.; writing—review and editing, A.F. and N.P.; visualization, N.C. and N.P.; supervision, A.F.; project administration, A.F. and N.P. All authors have read and agreed to the published version of the manuscript.

**Funding:** This research received no external funding.

**Conflicts of Interest:** The authors declare no conflict of interest.

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
