# Peer review of "Nano Silica and Metakaolin Effects on the Behavior of Concrete Containing Rubber Crumbs"

_2673-4109, doi:10.3390/civileng1030017_

Round 1
Reviewer 1 Report
A quiet comprehensive study is done and most of the parameters are considered.
- There are some papers published on the use of recycled rubber in concrete more recent (for example https://www.researchgate.net/project/ReCoTiP-Development-of-Reinforced-Concrete-Elements-and-Systems-with-Waste-Tire-Powder) that have not been taken into account. It is suggested to the authors to consider them.
- Please define the W/C ratio whether it is effective or not.
- Taking into account the small w/c ratio please explain how you prepared aggregate and how you have considered the amount of water needed to make the aggregate in a saturated surface dry state. Also, please indicate the total amount of water in Table 1.
- Line 101-there is: ultimate strength of 400kg/cm2? That should be a strength in MPa.
- Please specify Metakaolin as an % of sand and superplasticizer as an % of cement mass in Table 1, also % of rubber as a replacement for aggregate.
- It is hard to follow how did you define proportions and what are the differences between the composition of mixtures Please specify this more in detail.
- In general, it would be more interesting if the authors focus more on the significance of their findings regarding the importance of the interrelationship between the obtained results and sustainable development in the sector context, and the barriers to do it, what would be the consequences, in the real world, in changing the observed situation, what would be the ways, in the real world, to change/improve the observed situation. So, please write some remarks in the Conclusion about the significance of this research within the real sector.
- The English language used throughout the manuscript is good. In this study, original results and also, new insights into the topic were provided. The topic of this paper is suitable for the journal scope. Conclusions were related facts substantiated by this investigation.
Author Response
Nano silica and Metakaolin effects on the behavior of the concrete containing rubber crumbs
By Navid Chalangaran, Alireza Farzampour, Nima Paslar
Response to Reviewer Comments
Please find attached a second revised version of our manuscript “Nano silica and Metakaolin effects on the behavior of the concrete containing rubber crumbs”, which we would like to resubmit after major revision for publication as a Regular paper in CiviEng.
Your comments and those of the reviewers were highly insightful and enabled us to greatly improve the quality of our manuscript. In the following pages are our detailed and point-by-point responses to each of the comments of the reviewers as well as your own comments.
Revisions in the text are shown using yellow highlight for additions. In accordance with reviewers’ suggestions, we have revised following items.
We hope that the revisions in the manuscript and our accompanying responses will be sufficient to make our manuscript suitable for publication in CiviEng. We shall look forward to hearing from you at your earliest convenience.
Yours sincerely,
Alireza Farzampour, PhD, EIT
Dept. of Civil and Environmental Engineering
Virginia Tech
Blacksburg 22012, USA
Phone: +18144415272
E-mail: afarzam@vt.edu
A quiet comprehensive study is done and most of the parameters are considered.
- There are some papers published on the use of recycled rubber in concrete more recent (for example https://www.researchgate.net/project/ReCoTiP-Development-of-Reinforced-Concrete-Elements-and-Systems-with-Waste-Tire-Powder) that have not been taken into account. It is suggested to the authors to consider them.
Answer: The articles are added according to the reviewer’s suggestion.
- Please define the W/C ratio whether it is effective or not.
Answer: After several experiments, it was decided to select a water to cement ratio of 0.29 and all mixing designs were designed using a water to cement ratio of 0.29. The w/c ratio is effective on the mechanical properties of concrete, due post evaporation of the water, causing porosity in the concrete. This ratio was selected from two aspects, 1- reduction of porosity 2- reduction of the concrete flow. Crumb rubbers stay on the surface of the water, so the ratio of water to cement must be controlled so that the crumbs are fully integrated in all layers of concrete. The explanation is added to the manuscripts.
- Taking into account the small w/c ratio please explain how you prepared aggregate and how you have considered the amount of water needed to make the aggregate in a saturated surface dry state. Also, please indicate the total amount of water in Table 1.
Answer: Table 2 shows the weight of all materials in kilograms. As you know, the amount of water in concrete varies depending on the type of material and the amount of aggregate adsorption. On the other hand, when crumb rubber is used in the concrete, the flow of concrete increases due to lack of water absorption. The table was cluttered at the time of copying to the journal format, which has now been modified.
- Line 101-there is: ultimate strength of 400kg/cm2? That should be a strength in MPa
Answer: The articles are added according to the reviewer’s suggestion.
- Please specify Metakaolin as an % of sand and superplasticizer as an % of cement mass in Table 1, also % of rubber as a replacement for aggregate.
Answer: In the studied mixture, metakaolin and nano-silica are considered as adhesives (cement substitutes) and are measured based on the amount of cement and crumb rubber is replaced by sand. In N1%, N3% and N5% designs, due to the high water absorption of nano-silica, a superplasticizer has been used. In designs containing metakaolin and other designs, due to the desired psychological, there was no need to use superplasticizers. Superplasticizers was used with the aim of creating a favorable flow and slump of 50 to 70 mm.
- It is hard to follow how did you define proportions and what are the differences between the composition of mixtures Please specify this more in detail.
Answer: Based on the ACI 211 mixing design method, a general mixing design is designed and then modified using water and other material relations. Finally, after obtaining the exact number, the amount of material was partially changed using trial and error so that the concrete has a proper flow rate (50-70 mm slump). This concrete was considered as a control sample. Then, by replacing nanosilica (samples N1%, N3% and N5%) with cement, the optimal amount of nanosilica was calculated. Then, the optimal amount of metakaolin, rubber fine-grained and coarse-grained rubber was calculated and the optimal amounts of pozzolanic materials were combined with predefined percentages of rubber, and sub-rubber. (Table 2 was cluttered and modified accordingly).
This was the method of our mixing scheme:
Desired slump:
The largest aggregate diameter available:
Primary water:
Based on this method, according to the desired slump and the largest available aggregate diameter, from the tables in the standard, the initial water and water to cement ratio are interpolated.
Cement:
Apparent volume of Coarse sand: 0.62
Fine-grained sand Soft modulus: 2.8
The apparent volume of sand and the modulus of softness of sand at the time of granulation are calculated by placing in the following equations, the specific gravity of the aggregate, the density of fresh concrete, the weight of the sand and the weight of the dry sand.
Aggregate specific gravity:
Density of fresh concrete:
The amount of sand:
Weight of wet Coarse sand:
Weight of wet Fine-grained sand:
Water weight correction:
- In general, it would be more interesting if the authors focus more on the significance of their findings regarding the importance of the interrelationship between the obtained results and sustainable development in the sector context, and the barriers to do it, what would be the consequences, in the real world, in changing the observed situation, what would be the ways, in the real world, to change/improve the observed situation. So, please write some remarks in the Conclusion about the significance of this research within the real sector.
Answer: In recent decades, there was no tire recycling industry, and car tires, after being used and expired, were either buried in pits or after being set on fire, and disposing of this waste were considered as environmental pollution was ending. In the present study, due to the acceptable strength of the designed concrete, it is possible to use this concrete in the production of highway New Jersey blocks, heavy wall blocks, partition walls and other general concreting. Thus, crumb rubber is used as an aggregate without polluting the environment.
- The English language used throughout the manuscript is good. In this study, original results and also, new insights into the topic were provided. The topic of this paper is suitable for the journal scope. Conclusions were related facts substantiated by this investigation.
Answer: The authors would like to thank the reviewer for providing the valuable comments.

Reviewer 2 Report
The article entitled "Nano silica and metakaolin effects on the behavior of the concrete containing rubber crumbs" presents the benefits of increasingly using supplementary cementitious materials (SCMs) in composites with cement matrixes. This solution fits in with the subject of modern sustainable construction and therefore this manuscript is worth publishing. It is an interesting and original work, however, the paper suffers from some limits and needs significant improvements. The specific amendments are as follows:
(1) Besieds the cemical composition please provide also tha phase composition of cement and additions used.
(2) Figures with results should contain error bars. It is a necessity in the presentation of results of experimental studies without a thorough statistical analysis.
(3) Please provide more details about aggregate used in the studies, e.g. origin, mechnical parameters or chemical composition. The type of aggregate used has a significant impact on the many parameters of the composites.
(4) Please describe in more detail the test procedure of the samples for compressive strength, tensile strength etc., e.g. the loading speed of the samples, it was a static process?
(5) How accurately the data was collected. If there was no computer record of the research, how reliable are the results ? Please comment on this.
(6) Whether there were differences in the appearance of the samples after tests, i.e. after their destruction ? Please show example of samples or selected series of samples after performed tests.
(7) Section 5 should contain several clear concusions from the studies, given in points.
(8) This paper presents the potential of nano silica and metakaolin as superior additives for reinforcing concrete composites containing rubber crumbs. However, there are many other useful materials and binder substitutes that help to strengthen concretes with cement matrixes through synergy effect and make it more resistant to cracking from both static and dynamic loads, e.g. through reduction of interfacial microcracks between active pozzolanic additives and cement matrix. It is therefore required that the authors comment on the results of previous papers. In the Introduction section, the following new article from this topic should be discussed and cited:
Energies 2020, 13, 2184.
Author Response
Nano silica and Metakaolin effects on the behavior of the concrete containing rubber crumbs
By Navid Chalangaran, Alireza Farzampour, Nima Paslar
Response to Reviewer Comments
Please find attached a second revised version of our manuscript “Nano silica and Metakaolin effects on the behavior of the concrete containing rubber crumbs”, which we would like to resubmit after major revision for publication as a Regular paper in CiviEng.
Your comments and those of the reviewers were highly insightful and enabled us to greatly improve the quality of our manuscript. In the following pages are our detailed and point-by-point responses to each of the comments of the reviewers as well as your own comments.
Revisions in the text are shown using yellow highlight for additions. In accordance with reviewers’ suggestions, we have revised following items.
We hope that the revisions in the manuscript and our accompanying responses will be sufficient to make our manuscript suitable for publication in CiviEng. We shall look forward to hearing from you at your earliest convenience.
Yours sincerely,
Alireza Farzampour, PhD, EIT
Dept. of Civil and Environmental Engineering
Virginia Tech
Blacksburg 22012, USA
Phone: +18144415272
E-mail: afarzam@vt.edu
The article entitled "Nano silica and metakaolin effects on the behavior of the concrete containing rubber crumbs" presents the benefits of increasingly using supplementary cementitious materials (SCMs) in composites with cement matrixes. This solution fits in with the subject of modern sustainable construction and therefore this manuscript is worth publishing. It is an interesting and original work, however, the paper suffers from some limits and needs significant improvements. The specific amendments are as follows:
Answer: The authors would like to thank the reviewer for providing the valuable comments.
(1) Besieds the cemical composition please provide also tha phase composition of cement and additions used.
Answer: Water-cement ratio is considered to be 0.29, and the slump kept to be in the rage of 50±20 mm. Super plasticizer is initially mixed with water and poured into the mixer, then additives (Nano-silica or Metakaolin) are added to the mix. Subsequently, the aggregates and cement are poured into the mixer. For the sample with rubber crumbs, first the rubber crumbs and sand are mixed and then poured into the mixture following the producers for the rest of the mix designs. This discussion is added to the original manuscript following the reviewer’s comment.
(2) Figures with results should contain error bars. It is a necessity in the presentation of results of experimental studies without a thorough statistical analysis.
Answer: It was modified and figures are updated accordingly.
(3) Please provide more details about aggregate used in the studies, e.g. origin, mechnical parameters or chemical composition. The type of aggregate used has a significant impact on the many parameters of the composites.
Answer: Aggregate used from mines in the northwest of Fars province, Iran. The cement used is type 2 and it’s tone has been extracted and produced from mines in the northwest of Shiraz city in Fars province, Iran. Used nano-silica has been produced in Iran using silica from Yazd mines. Metakaolin is imported from China. The tire used was made from a car tire, which was crushed by an industrial shredder and then was classified.
(4) Please describe in more detail the test procedure of the samples for compressive strength, tensile strength etc., e.g. the loading speed of the samples, it was a static process?
Answer: The force is applied at a constant speed of 0.4 MPa. Samples of compressive strength are made in the form of cubes with dimensions of 15 * 15 * 15 cm. To calculate the compressive strength, the number read from the device is divided by the area of the sample surface and the compressive strength of the sample is obtained.
For tensile strength, a cylindrical specimen with dimensions of 15 x 30 cm has been used. This test examines the tensile strength by halving the cylindrical specimen. To calculate the tensile strength, after halving the sample, the number read from the device is placed in the following equation and the tensile strength is obtained.
The discussion is added to the original manuscript according to the reviewer’s comment.
(5) How accurately the data was collected. If there was no computer record of the research, how reliable are the results ? Please comment on this.
Answer: At the time of making the samples, 3 samples were made for each age. At the time of the experiment (application of load and failure of the sample) the results were read by two authors and written separately. It should be noted that all devices have been tested and calibrated on a monthly basis by the Standard Office of Iran.
(6) Whether there were differences in the appearance of the samples after tests, i.e. after their destruction ? Please show example of samples or selected series of samples after performed tests.
Answer: Samples without crumb rubber were broken 45 degrees and samples containing crumb rubber were broken in the similar way and no abnormal deformation was observed. All samples without crumbs were apparently broken:
Samples containing crumb rubber:
Samples without crumb rubber:
(7) Section 5 should contain several clear concusions from the studies, given in points.
Answer: Items were added following the review’s comment.
(8) This paper presents the potential of nano silica and metakaolin as superior additives for reinforcing concrete composites containing rubber crumbs. However, there are many other useful materials and binder substitutes that help to strengthen concretes with cement matrixes through synergy effect and make it more resistant to cracking from both static and dynamic loads, e.g. through reduction of interfacial microcracks between active pozzolanic additives and cement matrix. It is therefore required that the authors comment on the results of previous papers. In the Introduction section, the following new article from this topic should be discussed and cited:
Answer: In recent years, many pozzolanic materials have been proposed to improve the weaknesses and increase the strength of concrete. In the present study, we tried to compensate for the drop in strength of concrete containing crumb rubber by using the economical and most available materials. It is noted that if other materials were used, the cost of concrete containing crumb rubber would be much higher than the initial concrete and could not be used as a suitable industrial product. .

Round 2
Reviewer 2 Report
Unfortunately, the article was only partially improved. Still not achieved full response to some comments, e.g.
(1) The phase composition of mineral materials has not been provided.
(4) Loading rate of press should be given in MPa/s not in MPa.
(8) Reference to other new research results with additives is required because the article covers this topic. Therefore, it is legitimate to discuss and quote the following article from MDPI database.
Energies 2020, 13, 2184.
Based on the above an additional revision of the paper is required.
Author Response
Nano silica and Metakaolin effects on the behavior of the concrete containing rubber crumbs
By Navid Chalangaran, Alireza Farzampour, Nima Paslar
Response to Reviewer Comments
Please find attached a revised version of our manuscript “Nano silica and Metakaolin effects on the behavior of the concrete containing rubber crumbs”, which we would like to resubmit after major revision for publication as a Regular paper in CiviEng.
Your comments and those of the reviewers were highly insightful and enabled us to greatly improve the quality of our manuscript. In the following pages are our detailed and point-by-point responses to each of the comments of the reviewers as well as your own comments.
Revisions in the text are shown using yellow highlight for additions. In accordance with reviewers’ suggestions, we have revised following items.
We hope that the revisions in the manuscript and our accompanying responses will be sufficient to make our manuscript suitable for publication in CiviEng. We shall look forward to hearing from you at your earliest convenience.
Yours sincerely,
Alireza Farzampour, PhD, EIT
Dept. of Civil and Environmental Engineering
Virginia Tech
Blacksburg 22012, USA
Phone: +18144415272
E-mail: afarzam@vt.edu
.
(1) The phase composition of mineral materials has not been provided.
Answer: The explanations are added to the manuscript and highlighted following the reviewer’s comment,
(4) Loading rate of press should be given in MPa/s not in MPa.
Answer: It is revised accordingly.
(8) Reference to other new research results with additives is required because the article covers this topic. Therefore, it is legitimate to discuss and quote the following article from MDPI database.
Answer: It is added and discussed accordingly.
The authors would like to thank the reviewer for providing the valuable comments

Round 3
Reviewer 2 Report
I have no comments.